# Early feasibility study with an implantable near-infrared spectroscopy sensor for glucose, ketones, lactate and ethanol

**Francesca De Ridder** [1,2], **Rie Braspenning**[1], **Juan S. Ordonez**[3], **Gijs Klarenbeek**[3], **Patrick Lauwers**[4], **Kristien J. Ledeganck**[2], **Danaë Delbeke**[3], **Christophe De Block** [1,2]*

1 Department of Endocrinology-Diabetology-Metabolism, Antwerp University Hospital, Antwerp, Belgium,
2 Faculty of Medicine & Health Science, Laboratory of Experimental Medicine and Pediatrics, University of Antwerp, Antwerp, Belgium, 3 Indigo Diabetes N.V., Ghent, Belgium, 4 Department of Vascular & Thoracic Surgery, Antwerp University Hospital, Antwerp, Belgium

* christophe.deblock@uza.be, christophe.deblock@uantwerpen.be

## Abstract

### Objective

To evaluate the safety and performance of an implantable near-infrared (NIR) spectroscopy sensor for multi-metabolite monitoring of glucose, ketones, lactate, and ethanol.

### Research design and methods

This is an early feasibility study (GLOW, NCT04782934) including 7 participants (4 with type 1 diabetes (T1D), 3 healthy volunteers) in whom the YANG NIR spectroscopy sensor (Indigo) was implanted for 28 days. Metabolic challenges were used to vary glucose levels (40–400 mg/dL, 2.2–22.2 mmol/L) and/or induce increases in ketones (ketone drink, up to 3.5 mM), lactate (exercise bike, up to 13 mM) and ethanol (4–8 alcoholic beverages, 40-80g). NIR spectra for glucose, ketones, lactate, and ethanol levels analyzed with partial least squares regression were compared with blood values for glucose (Biosen EKF), ketones and lactate (GlucoMen LX Plus), and breath ethanol levels (ACE II Breathalyzer). The effect of potential confounders on glucose measurements (paracetamol, aspartame, acetylsalicylic acid, ibuprofen, sorbitol, caffeine, fructose, vitamin C) was investigated in T1D participants.

### Results

The implanted YANG sensor was safe and well tolerated and did not cause any infectious or wound healing complications. Six out 7 sensors remained fully operational over the entire study period. Glucose measurements were sufficiently accurate (overall mean absolute (relative) difference MARD of 7.4%, MAD 8.8 mg/dl) without significant impact of confounders. MAD values were 0.12 mM for ketones, 0.16 mM for lactate, and 0.18 mM for ethanol.

**Data Availability Statement:** The authors will make the study data sets available upon reasonable request. The request can be addressed to Prof. Dr.

Christophe De Block, PI of the study. Contact: e-mail: christophe.deblock@uza.be Address: Antwerp University Hospital, Drie Eikenstraat 655, 2650 Edegem, Belgium Alternatively, requests for data may be directed to the following non-author recipient: Indigo Diabetes NV e-mail: info@indigomed.com Address: Bollebergen 2B, Box 5, 9052 Gent, Belgium.

**Funding:** The study was funded by Indigo Diabetes NV. Danaë Delbeke and Juan Ordonez are employees of Indigo Diabetes NV. FDR received a Strategic Basic Research Grant (1S62821N) from Research Foundation – Flanders (FWO). As sponsor of the study, Indigo Diabetes was involved in trial design (writing of study documents, such as the study protocol), trial execution (contracted to QbD Clinical, a CRO based in Belgium) and analysis of trial results.

**Competing interests:** I have read the journal's policy and the authors of this manuscript have the following competing interests: Christophe De Block: consulting fees and honoraria for speaking for Abbott, AstraZeneca, Boehringer-Ingelheim, A. Menarini Diagnostics, Eli Lilly, Medtronic, Novo Nordisk, and Roche, and research support from AstraZeneca, Boehringer-Ingelheim, Indigo Diabetes and Novo Nordisk. Gijs Klarenbeek (Melfin Medical Consulting) has a consultancy agreement with Indigo Diabetes. Danaë Delbeke and Juan Ordonez are employees of Indigo Diabetes NV. All authors read the journal's policy on competing interests. This does not alter our adherence to PLOS ONE policies on sharing data and materials.

## Conclusions

The first implantable multi-biomarker sensor was shown to be well tolerated and produce accurate measurements of glucose, ketones, lactate, and ethanol.

## Trial registration

Clinical trial identifier: NCT04782934.

## Introduction

Reaching and maintaining optimal glycemic control while avoiding both hypo- and hyperglycemia remains a daily challenge for people living with type 1 diabetes (T1D), despite advances in insulin therapy and glucose monitoring options [1]. The introduction of continuous glucose monitoring (CGM) has changed management of T1D and, in comparison with self-monitoring of blood glucose, it is associated with less severe hypoglycemia, higher treatment satisfaction and less work absenteeism, with comparable or lower HbA1c values [2–5].

CGM systems use a wearable needle-type sensor to measure interstitial glucose using a glucose oxidase based enzymatic method. Whereas intermittently scanned CGM (isCGM) devices only report glucose values and trends on demand when the wearer scans the sensor, real-time CGM (rtCGM) systems automatically transmit glucose values to a receiver every 1–5 minutes and can provide predictive alerts warning the user of impending hypo- or hyperglycemia [6], thereby increasing time in range (70–180 mg/dl or 3.9–10 mmol/l) and improving HbA1c values [7,8]. However, limitations of the currently available CGM systems include a 7–14 day sensor lifetime limit, sensor loss, visibility of the sensor [9]. Up to 5.5% of patients develop adverse skin reactions, mainly contact dermatitis in reaction to isobornyl acrylate in the sensor adhesive [10,11].

An implantable CGM system would overcome a number of these limitations. The Eversense system is currently the only implantable CGM system on the market. It uses a fluorescence-based sensor with a life span of 90–180 days in combination with a wearable transmitter and a smartphone application [12]. While implantable sensors may offer benefits for people with skin problems or oversensitivity to adhesives, and for those with dexterity impairment or those who want the option of transient removal of external devices (privacy, professional reasons), monitoring additional biomarkers to go from CGM to continuous multi-biomarker monitoring (CMM) is expected to bring additional advantages.

Continuous ketone monitoring (CKM) in addition to glucose would provide a clear benefit for people with type 1 diabetes, as diabetic ketoacidosis (DKA) is an important and potentially life-threatening complication still occurring in 10–130 per 1000 person-years due to a variety of situations, including insufficient insulin administration due to insulin pump or pen malfunction, inadequate bolus dosing, or during sick days, high-intensity exercise, hyperemesis gravidarum, excessive alcohol intake or cannabis use [13–15]. Also, CKM would help to optimize insulin delivery for people on a very low carbohydrate diet or using sodium-glucose cotransporter-2 inhibitors (SGLT-2i).

In practice, recommended monitoring for ketoacidosis is often not performed in a timely manner [16]. Moreover, only few patients own home ketone monitoring strips, and in one third of cases these strips have exceeded their expiry date [17]. More importantly, ketone strip testing needs to be initiated by the user and thus cannot provide predictive trend information,

whereas a multianalyte platform with continuous ketone monitoring can provide a timely warning or alert for impending ketoacidosis, allowing remedial action to be taken [13,14,18–20].

Physical activity is a cornerstone of a healthy lifestyle but induces rapid fluctuations in glycemia in individuals with T1D due to glucose consumption by active muscles, and changes in counter-regulatory hormone levels. Insufficient information and knowledge to manage exercise-induced changes in glycemic control and fear of hypoglycemia can make individuals with T1D reluctant to or refrain from intense or prolonged physical activity [21–23].

Continuous lactate monitoring (CLM) can be useful to allow persons with diabetes to better judge the impact of their exercise regimen on glycemic control and could be used to adjust insulin administration during exercise and alert people to severe physiological stress [24]. Indeed, lactate increases with anaerobic exercise, which, in contrast to aerobic exercise, does not cause hypoglycemia during exercise [23]. Furthermore, CLM may help T1D athletes to achieve their exercise goals more effectively while remaining in optimal glucose range by keeping lactate levels below the lactate threshold. The lactate threshold, which is approximately 2 mmol/L, is the level above which lactate accumulates faster in the blood than it can be removed. As such, it is the cut-off point for exercise intensity that cannot be sustained, the most reliable predictor for endurance performance [25]. A minority of people with T1D uses metformin as adjunctive therapy and should be aware of the potential danger of lactic acidosis with acute kidney injury [26].

People with T1D should also be aware of the effects of alcohol consumption as it increases the risk of hypoglycemia, mainly by inhibiting gluconeogenesis, tempering the counter-regulatory response and contributing to reduced hypoglycemia awareness [27].

Here we describe an early feasibility clinical study with the YANG sensor, an investigational implantable device for continuous multi-metabolite monitoring of glucose, ketones, lactate, and ethanol using near-infrared (NIR) spectroscopy technology. Preclinical data with this technology in a Göttingen minipig model have shown that continuous and simultaneous measurement of glucose, beta-hydroxybutyrate and lactate over a 2-month period was feasible both in and outside the physiological range with low between-sensor accuracy variation, and that levels of lactate and beta-hydroxybutyrate in the interstitial fluid closely correlate to blood levels [28]. As NIR does not require any reagents that become depleted or unstable, the sensor is expected to reach a life span of at least two years.

## Research design and methods

### Study characteristics

The GLOW study (ClinicalTrials.gov identifier NCT04782934), a monocentric open label early feasibility, first-in-human study with the implantable YANG sensor for continuous monitoring of glucose, ketones, lactate and ethanol in people with type 1 diabetes and healthy volunteers, was conducted at Antwerp University Hospital between February 25 and July 19, 2021. The study included 7 participants, 4 with type 1 diabetes and 3 healthy volunteers.

The study design is summarized in Fig 1. Between sensor implantation (V1) and explantation (V8) 28 days later, study participants underwent 6 measurement visits (V2-V7) in which glucose, ketones, and lactate were measured and the effect of various metabolic challenges and confounders on the measurements was evaluated. Two follow-up visits were conducted after explantation: one for wound inspection and removal of sutures 10 days after explantation (V9) and a follow-up phone call after 4 weeks (V10). Photographs of healed implantation sites were taken after 4–6 weeks.

The study was approved by the ethics committee of Antwerp University Hospital (EC Belgian approval number B3002020000190, UZA nr 20/40/530) and compliant with the

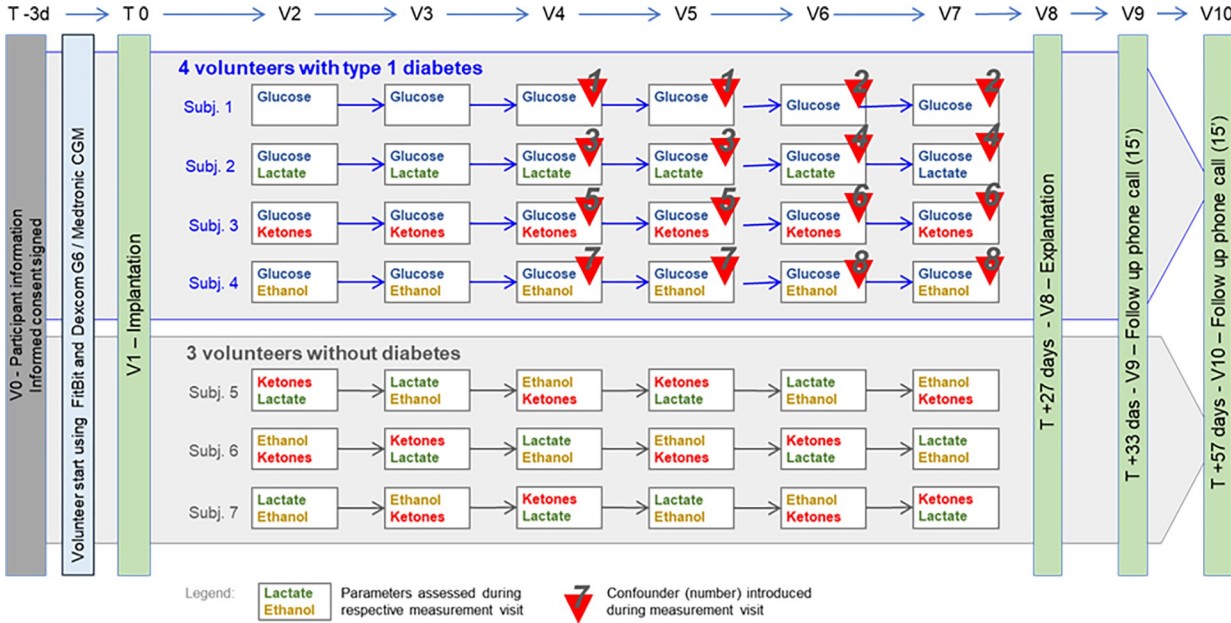

**Fig 1. Study setup.** Visits. V0: Inclusion & screening, V1: Implantation of the YANG sensor, V2-V7: Measurement visits, during which study participants were subjected to metabolic challenges (indicated in green, yellow or red) and effect of confounder compounds on glucose measurements were assessed in participants with type 1 diabetes (numbered red triangles); V8: Explantation, V9: Suture removal and wound healing follow-up; V10: Follow-up phone call. Confounding compounds and doses. The serial number in this list corresponds to the number in the red triangles on the study setup scheme: 1. Paracetamol (1000 mg p.o.), 2. aspartame (2 g p.o.), 3. acetylsalicylic acid (1000 mg p.o.), 4. ibuprofen (1000 mg p.o.), 5. sorbitol (20 g p.o.), 6. caffeine (4 espressos, ca. 260 mg p.o.), 7. fructose (50 g p.o.), 8. vitamin C (2 g p.o.).

Declaration of Helsinki and the International Conference on Harmonization/Good Clinical Practice guidelines. All participants gave written informed consent before entering the study.

Inclusion criteria for this study were: adults up to 50 years of age with body mass index (BMI) between 20 and 27.5 kg/m$^2$. Volunteers with type 1 diabetes needed to fulfill the WHO diagnostic criteria and using an insulin pump for at least 12 months. Healthy volunteers were required to be free of chronic disease, have a normal physical examination at screening and be able and willing to perform intense physical activity during measurement visits. General exclusion criteria were coagulation or bleeding disorders or taking anticoagulant medication. Additional exclusion criteria were severe hypoglycemia or diabetic ketoacidosis requiring emergency room visit or hospitalization in the past 6 months for participants with type 1 diabetes and impaired fasting glucose or impaired glucose tolerance for healthy volunteers.

## Objectives and endpoints

The primary endpoints were the safety and biocompatibility of the YANG sensor and the incidence of sensor failure. Safety data comprised device related events as well as procedure related adverse events linked to the insertion or removal of the YANG sensor. The biocompatibility evaluation included histological assessment of foreign body reactions to sensor implantation.

Secondary endpoints were implantation and explantation procedure parameters (user feedback, resources needed and duration) and post-explantation follow-up data.

The sensor's ability to measure glucose, β-hydroxybutyrate, lactate and ethanol levels and the effect of potentially interfering substances (paracetamol 1000 mg, aspartame 2 g, acetylsalicylic acid 1000 mg, ibuprofen 1000 mg, sorbitol 20 g, caffeine from 4 espressos (about 260 mg), fructose 50 g, vitamin C 2 g) on these measurements were explorative endpoints for this study.

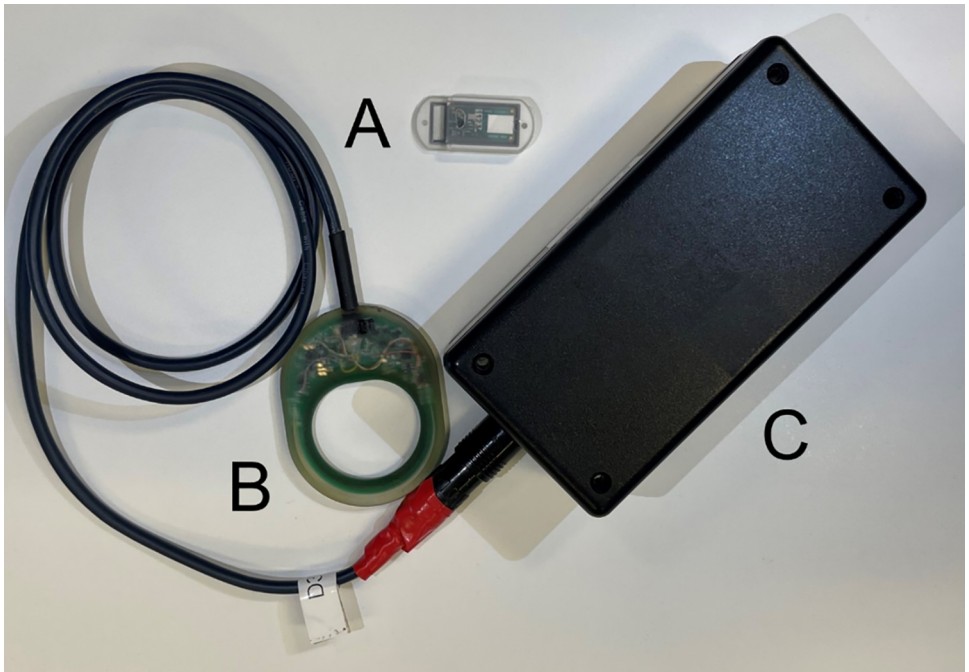

**Fig 2. Components of the YANG system for continuous metabolic monitoring.** The YANG system consists of an implantable sensor of size 3 x 0.7 x 1.5 cm (A) and a charging and communication antenna (B) connected to a data storage box (C), forming the YANG external device (YANGEXD). During measurement visits, the sensor was wirelessly charged via the charging and communication antenna applied to the skin over the sensor using an adhesive patch.

## Device design

The YANG system for continuous metabolic monitoring consists of the subcutaneously implanted YANG sensor, a charging and communication antenna and a data storage box (Fig 2). The device used in this study is an investigational prototype, subject to a hardware update prior to clinical use.

The YANG sensor is a miniaturized near-infrared spectrometer on a silicon photonics chip that measures optical transmittance in the interstitial fluid at up to 24 wavelengths between 1680 and 2400 nm. Its optical components and electrical assemblies are mounted on a silicon-based substrate, enclosed by a glass capsule, reinforced by an alumina helmet for mechanical protection and covered by a biocompatible silicone envelope. It further contains a small coil for wireless communication and charging, and a micro battery. The size of the sensor is 3 cm x 0,7 cm x 1,5 cm (3.15 ml), mainly taken up by the size of its internal battery.

## Implantation

The sensor was implanted under local anesthesia by a surgeon in an outpatient setting. After local anesthesia with 1% lidocaine, a vertical 35 mm incision was made in the left lower quadrant of the abdomen. The location was chosen as caudal as possible and in consultation with the participant. After incision, a subcutaneous pocket was created just large enough to accommodate the device and the sensor was inserted with its sensing area facing down. Depending on the amount of subcutaneous fat, the sensor was implanted 5–10 mm below the skin surface. The sensor was not secured in the subcutaneous pocket with stitches. The skin incision was

closed in two layers using absorbable sutures (Vicryl 2/0) for the subcutaneous layer and non-absorbable sutures (Monocryl 3/0) for the skin. Finally, a small bandage was applied.

## Measurement visits

Over the 28-day implantation period, study participants were subjected to different metabolic challenges in six 8-hour long measurement visits (V2-V7) during which the levels of glucose, ketones (beta-hydroxybutyrate), lactate and ethanol measured via the YANG sensor were compared to those in blood samples taken at 5-minute intervals (2.5-minute interval when glucose levels <70 mg/dl or 3.9 mmol/l). Glucose levels were additionally compared to those measured in interstitial fluid using the Dexcom G6 CGM system. In participants with type 1 diabetes, the effect on glucose measurements of lactate, ketones, and ethanol alone or in combination with potential trends of simultaneously captured metabolites without (on measurement visits 1 and 2) and with potential confounding agents (on measurement visits 3–6) was investigated. In healthy volunteers, the effect of a combination of two metabolic challenges (ketones+lactate, ethanol+ketones and lactate+ethanol) was investigated, as detailed in Fig 1.

Except during measurement visits, the YANG sensor was in the off state. At the start of each measurement visit, the external device was brought in close proximity to the sensor and the system was allowed to equilibrate before starting the measurements.

Participants with t*ype 1 diabetes.* In each of the measurement visits of participants with type 1 diabetes, controlled hyperglycemia and hypoglycemia were induced sequentially. Hyperglycemia was induced by stopping the insulin pump and administration of a glucose drink containing 100 g of glucose (Glucomedics Lemon 100g, Lambra, Madrid, Spain) up to a glucose level of 400 mg/dl. To induce hypoglycemia, the insulin pump was restarted, and insulin was given using the subject's personal insulin correction factor down to blood glucose values of 40 mg/dl, with a maximum decrease rate of 100 mg/dl/h. The insulin pump was then switched off again until reaching normoglycemia.

In subject 1, no additional metabolic challenge was added to the controlled hyper- and hypoglycemia, whereas the other participants with type 1 diabetes underwent an additional metabolic challenge, with resp. lactate, ketones or ethanol. Subject 2 was subjected to intensive physical activity (stationary bicycle) to raise lactate levels to a peak level of 13 mM. In subject 3 ketone levels were increased by a low carbohydrate diet (3 days before measurement 1 and during measurement days 3 and 4), supplemented with oral administration of a ketone ester drink (HVMN Ketone ester up to 330 mg/kg body weight) to reach ketone levels of at least 3.5 mM. Subject 4 alternated moderate ethanol (3–4 alcoholic beverages, corresponding to 30–40 g of ethanol) and high ethanol (6–8 alcoholic beverages, corresponding to 60–80 g of ethanol) measurement visits, targeting a maximal ethanol level of around 20 mM (measured every 10 minutes with a breath analyzer [Digital Breathalyzer ACE II Basic plus, ACE Instruments, Freilassing, Germany]), with ethanol intake starting at least 1 hour after measurement start.

After 2 measurement visits to establish a baseline, every subject received 2 different confounding compounds, each one repeated on two subsequent visits (Fig 1). Confounder compounds were administered per os in the doses indicated in Fig 1.

Healthy volunteers. In healthy volunteers, the levels and course of ketones, lactate, and ethanol on glucose measurements with the YANG sensor were evaluated by measuring the effect of different combinations of two of these interferents in one visit. In every subject, three different combinations were evaluated twice each. The first intervention started one hour after the start of the measurement visit, with the second intervention starting at least 3 hours later.

In the ketones + lactate challenge, a ketone ester drink (HVMN, Miami, Florida) containing 30g of ketones was administered, followed by exercising on a stationary bicycle for up to 2 hours to reach lactate levels of up to 13 mM.

In the combined ethanol + ketones challenge, oral ethanol intake (6–8 alcoholic beverages, corresponding to 60–80 g of ethanol–to achieve blood ethanol levels of approximately 20 mM) was followed by ketone ester drink containing 30 g of ketones.

The lactate + ethanol challenge was achieved by exercising on stationary bicycle for up to 2 hours to reach lactate levels of up to 13 m, followed by intake of alcoholic beverages to reach blood alcohol levels of approximately 20 mM.

## Blood sample analyses

Blood samples were analyzed for glucose with the EKF Diagnostics–Biosen C-Line Glucose and Lactate Analyzer and with glucose strips GlucoMen LX Plus Sensor, read on GlucoMen LX2 reader (measuring range: 0.5–50 mmol/l, CV $\leq$1.5%), lactate was measured with the same analyzer using the EKF Diagnostics–Biosen C-Line Glucose and Lactate Analyzer (measuring range: 0.5–40 mmol/l, CV $\leq$1.5%), and ketone levels were measured with ketone test strips GlucoMen LX B(eta)-Ketone Sensor with reader GlucoMen LX2 (measuring range 0.1–8.0 mmol/l, CV 10.1–3.0%).

## Explantation procedure

Ultrasound imaging of the implant site prior to removal of the implant was performed in 4 subjects. The explantation procedure was performed as an outpatient procedure. After disinfection and sterile draping of the abdomen, local anesthesia using up to 10 ml of lidocaine was administered. The implantation pocket was opened using at the location of the implantation scar. The minimal capsule that had formed around the sensor was opened, the device was grasped carefully with a small Kocher clamp and extracted. A small biopsy of the proximal capsule was taken, and a biopsy of the distal implantation pocket was taken if possible. The subcutaneous pocket was rinsed with isobetadine before closing the incision in two layer, with: resorbable stitches for the subcutaneous layer (Vicryl 2/0), and an intradermal suture was applied for skin closure (Monocryl 3/0). A small bandage was applied. Standard wound care was performed until removal of sutures after 10 days.

## Histological analysis

Explantation biopsies were fixed in formalin and embedded in paraffin. Hematoxylin and eosin (HE) stained sections were used for routine diagnostic evaluation by a single pathologist.

## Data analysis

Adverse event and device failure incidence, as well categorical data from questionnaire responses are presented as frequencies or percentages. Continuous variables are presented as mean ± standard deviation or quartiles.

Partial least squares (PLS) regression analysis was used to link YANG sensor optical attenuated total reflection (ATR) transmission spectra to the relevant metabolites (glucose, β-hydroxybutyrate, lactate or ethanol). To assess the robustness and accuracy of the model during the clinical visits and evaluate the effect of confounding compounds on the YANG sensor measurements, the PLS model was trained each day and contiguous blocks cross-validation was used to estimate its performance in practice. In the contiguous blocks approach, 8 consecutive blocks of 12–15 data pairs per block were defined, of which one block was used for

validation. The root-mean-square error of cross validation (RMSECV) was calculated for the independent blocks as a measure of the difference between observed and expected values, and statistically evaluated with a t-test (with significance accepted at p<0.05). The cross-validation data is presented in scatter plots of reference measurements and YANG measurements. For glucose, the Parkes Error Grid was used to evaluate measurement accuracy in line with current standards for clinical use.

Accuracy for sensor glucose, ketones, lactate, and ethanol, in comparison with values measured in blood as reference, is expressed as mean absolute difference (MAD) and mean absolute relative difference (MARD).

## Results

Study population characteristics and baseline measurements are summarized in Table 1. The mean age was 30 ± 7 years and mean BMI 24.3 ± 2.5 kg/m$^2$. Every subject was followed for the complete duration of the study and completed the 4-week implantation period.

### Safety and adverse events

The YANG sensor was safe and well tolerated and did not give rise to any infections or wound healing complications. A limited hematoma was observed at the implantation site in two participants.

Ultrasound imaging before explantation was performed in 4 patients and showed minimal to no encapsulation around the implant, with no to minimal fluid collection around the device. Implanted sensors were located between 3 and 10 mm below the skin (Fig 3).

Histological analysis of explantation biopsies revealed mild reactive changes. A giant cell foreign body reaction to the suture material was observed in one participant with type 1 diabetes (Fig 4).

Implantation and explantation wounds healed without excessive scar formation, as shown in the photographs of healed scars taken after 4–6 weeks (Fig 5).

### Implantation & explantation procedures

Since this was a first-in-human study with this implantable device, the implantation was performed by a single surgeon. The surgeon was trained on the implantation and explantation procedures by classroom and animal model training and found both procedures easy to perform. The total implantation procedure lasted 15 ± 2 minutes. There were no issues locating or retrieving the sensors at the time of explantation.

**Table 1. Population characteristics.** Results are expressed as mean ± SD for continuous variables and as n (%) for categorical variables.

|  | All | T1D | Healthy controls |
|---|---|---|---|
| Number of patients | 7 | 4 | 3 |
| Age (years) | 30±7 | 29±8 | 31±5 |
| Gender (Male) | 3 (43%) | 1 (25%) | 2 (67%) |
| Blood pressure (mm Hg) | 119±9 / 76±8 | 120±9 / 77±8 | 117±9 / 73±8 |
| BMI (kg/m2) | 24.3±2.5 | 24.9±1.7 | 23.6±3.2 |
| Time since diagnosis (years) | - | 8.8±5.4 | - |
| HbA1c (mmol/mol) | - | 55±6 | 35±4 |

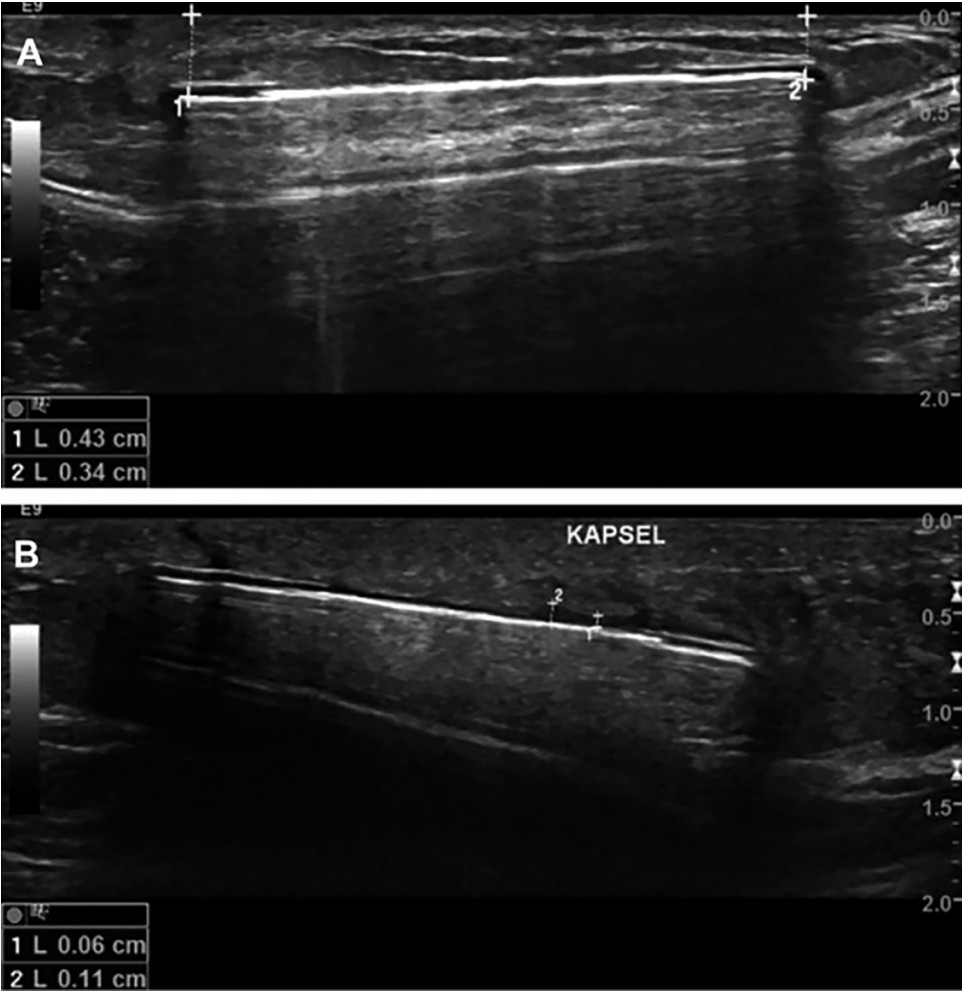

**Fig 3. Ultrasound images of sensor location.** Representative ultrasound images (max: 5 days before explantation) showing sensor implantation depth and minimal encapsulation and fluid collection around the sensor. A: Healthy control, sensor implantation depth of 0.34 to 0.43 cm below the skin surface. B: T1D: Minimal encapsulation and fluid collection (0.06 to 0.11 cm) around the sensor.

### User experience

The implantation procedure was rated as comfortable by six subjects and somewhat uncomfortable by one subject. Most study participants did not experience any inconveniences related to the implanted Yang sensor and did not feel limited in professional or leisure activities. Interference of the implant with normal sleep habits, the implant being somewhat visible and interference of the implant with sports activities were reported by one participant each.

### Sensor performance

Six out of seven YANG sensor implants remained operational during the full 28-day implantation period. For one sensor, the communication with the external device was lost during the third measurement visit (V4) and could not be re-established. Further measurement visits for this subject (subject 7) were cancelled.

Over the course of the study, a connection or communication issue between the sensor and the external charging and communication device occurred 15 times, mostly due to imprecise

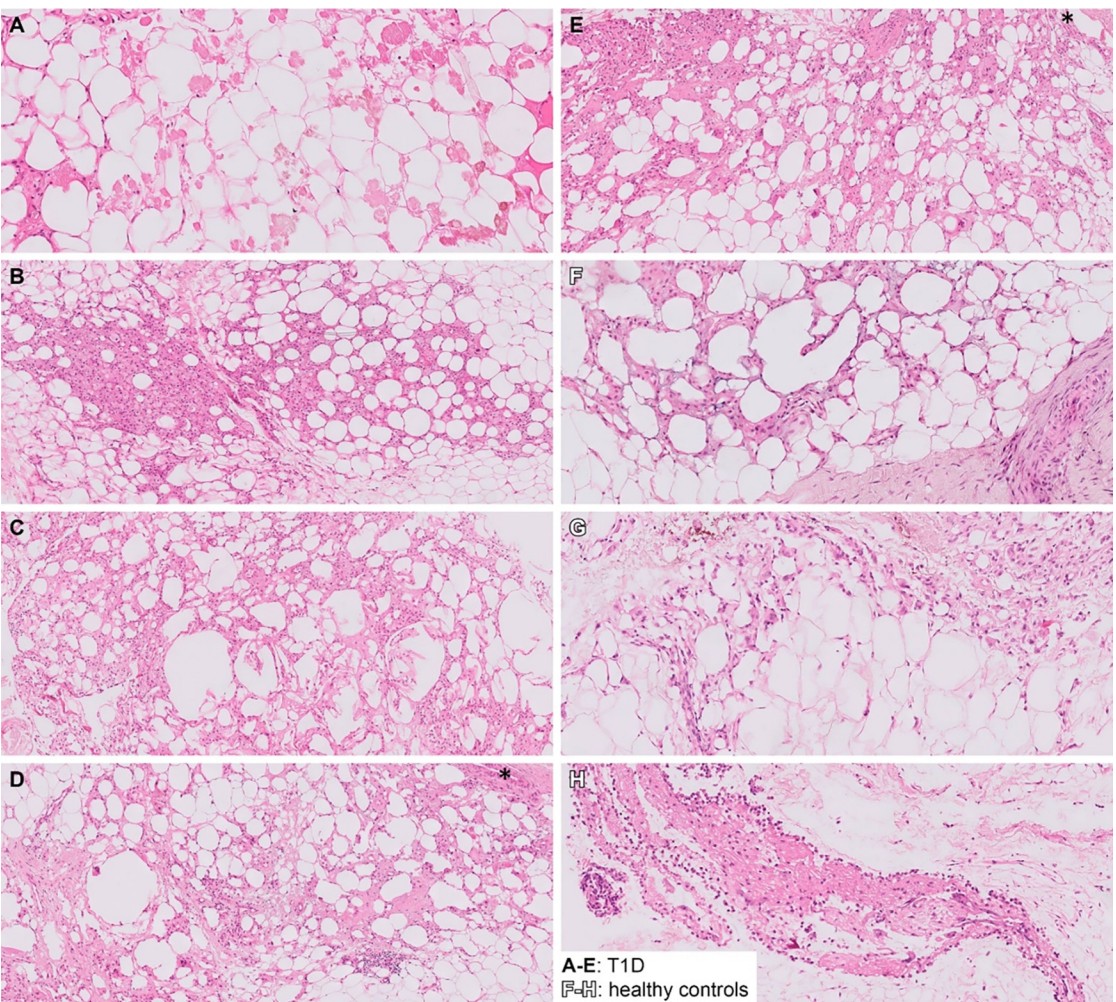

**Fig 4. Histology of explantation biopsies.** HE staining of explantation biopsies showing mild reactive changes. A-E: Biopsies from T1D participants, F-H: Biopsies from healthy volunteers *: Photomicrographs of the same patient showing foreign body reaction with giant cells, possibly in reaction to the suture material.

placement of the external device over the sensor, once because the device was damaged by being dropped from a height. These issues were resolved during the measurement visit by realigning or replacing the device.

## Explorative endpoints: Measurement of glucose, lactate, ketones, and ethanol

Measurements obtained with the YANG sensor were compared with Biosen EKF (1988 samples) and Dexcom G6 CGM (n = 2513) for glucose, with Biosen EKF for lactate (n = 1804), with ketone strips (n = 635) and with breath analyzer measurements for ethanol (n = 268).

For the participants with type 1 diabetes, glucose concentrations ranged from values as low as 40 mg/dl up to values above 300 mg/dl and displayed bidirectional rates of change up to 4 mg/dl/min. Fig 6A–6F shows glucose values obtained by the YANG sensor in comparison with Biosen EKF in function of time for all measurements visits of a representative participant with type 1 diabetes (subject 3).

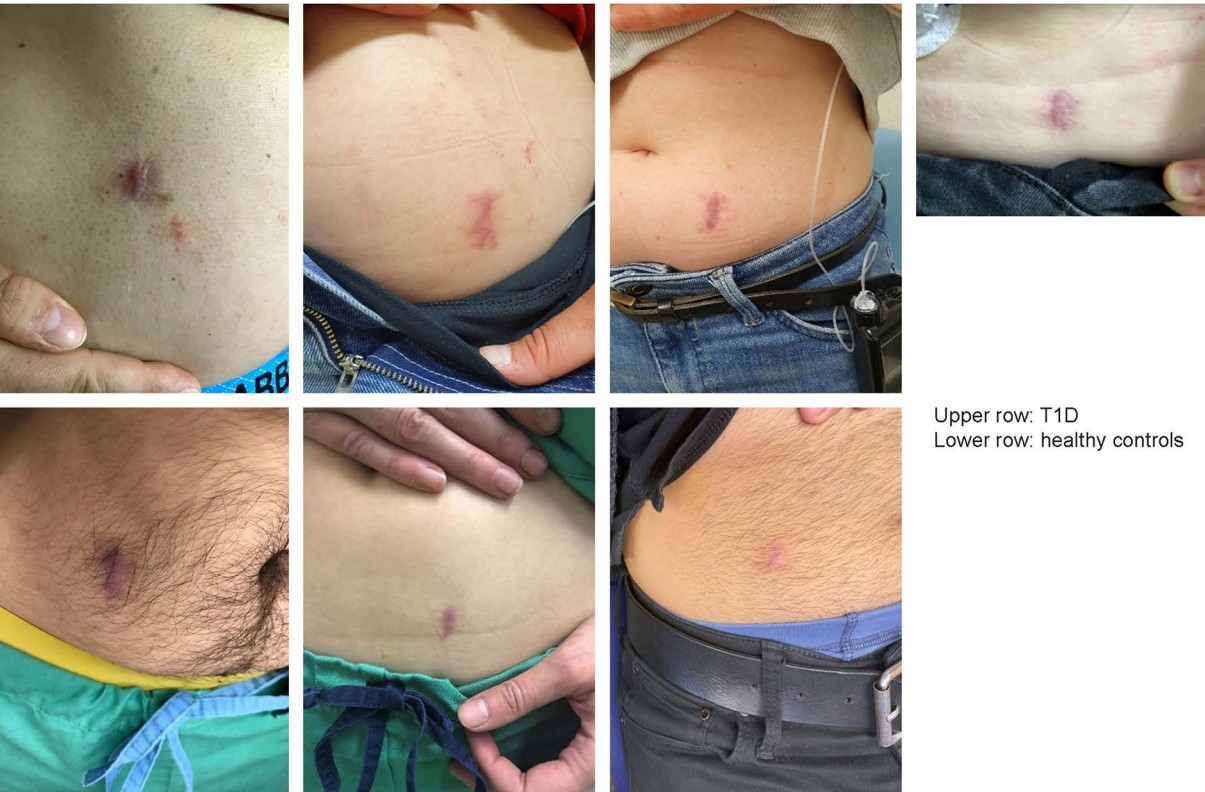

Upper row: T1D
Lower row: healthy controls

**Fig 5. Pictures of healed implantation sites.** Implantation site 4–6 week after explantation of the YANG sensor for all participants. Upper row: Subjects with type 1 diabetes. Lower row: Healthy volunteers.

The Parkes error grid in Fig 6I aggregates all glucose measurements according to the PLS cross validation model of NIR spectra compared to the Biosen EKF reference measurement of the 4 participants with type 1 diabetes obtained throughout the study, including measurements in the presence of confounders. The quality of the optical spectra did not change over the implantation period. For glucose measurements with the YANG sensor, a MAD value of 6.0 mg/dl and MARD of 11.8% were obtained for glucose values between 40 and 70 mg/dl. In the euglycemic range of 70–180 mg/dl, MAD was 7.5 mg/dl and MARD 6.5%, whereas for hyperglycemic conditions above 180 mg/dl, MAD was 7.2 mg/dl and MARD 2.9%. Overall, 98.9% of all data points on the Parkes error grid were in zone A, 1.1% in zone B. The overall MARD was 7.1%. On the Change Rate Error Grid (Fig 6J), 93.4% of the measurements were in zones A+B.

Administration of paracetamol, acetylsalicylic acid, ibuprofen, sorbitol, caffeine, fructose, aspartame, and vitamin C did not significantly influence the accuracy of glucose measurement by the YANG sensor. Fig 6H presents the calculated residuals for glucose measurement in the presence of confounders acetylsalicylic acid and ibuprofen (subject 3). A p-value $< 0.05$, implies that the confidence interval for the difference between the means falls within the pre-specified threshold of 10.5 mg/dl–a threshold set according the iCGM criteria to determine a non-significant delta. This indicates that consuming the confounder in large quantities, will not critically impact the glucose accuracy.

Continuous measurement of ketones was compared to blood strip measurements in the range of 0 to 4 mM as shown in Fig 6K. Ketone concentrations were determined by the sensor with an accuracy of 0.12 mM. The lactate concentrations measured over a range of 0 mM– 20

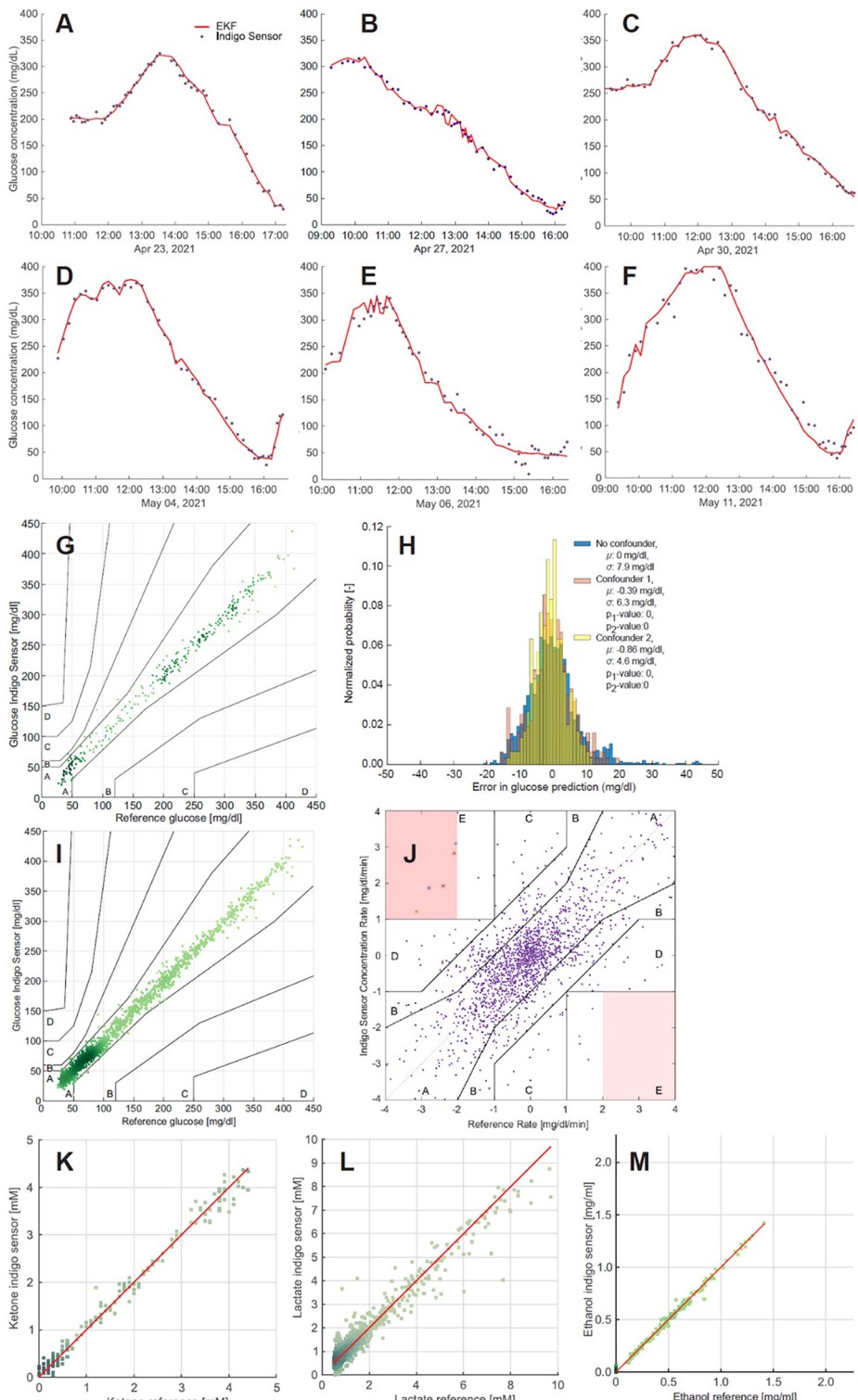

**Fig 6. Sensor measurements. A-H: Yang sensor for a representative patient with type 1 diabetes (subject 3). A-F:** Glucose measurements with YANG sensor versus EKF in function of time for all measurement visits. PLS regression cross-validation values of the YANG sensor versus EKF reference data in function of time for all measurement visits of subject 3. Glucose was measured in the presence of ketones (A-C, E) and in the presence of sorbitol (C,D) and caffeine (E,F) as potential confounders. **G**: Parkes error grid of glucose measurements with YANG sensor versus EKF

combined for all measurement visits. Overall accuracy: MAD 8.8 mg/dl, MARD 7.4%. In glucose range 40–70 mg/dl: MAD 9.2 mg/dl, MARD 18.2%; range 70–180 mg/dl: MAD 9.9 mg/dl, MARD 9.2%; >180 mg/dl: MAD 8.6 mg/dl, MARD 3.2%. **H:** Impact of confounders on glucose measurements. Histogram of the error in glucose prediction of the YANG sensor versus Dexcom for measurements without confounders (blue bars) and in the presence of sorbitol (confounder 1, pink bars) and caffeine (confounder 2, yellow bars). **I-J: Glucose measurements for all subjects with type 1 diabetes I:** Parkes error grid for glucose measurements with YANG sensor versus Biosen EKF. Aggregated data for all subjects with type 1 diabetes over all measurement visits, including measurements in the presence of potential confounders. Range 40–70 mg/dl: MAD 6.0 mg/dl, MARD 11.8%; range 70–180 mg/dl: MAD 7.5 mg/dl, MARD 6.5%; >180 mg/dl: MAD 7.2 mg/dl, MARD 7.1%. **J:** Error grid analysis for glucose measurement with YANG sensor versus EKF. Aggregated data for all subjects with type 1 diabetes and all measurement visits, including measurements in the presence of potential confounders. 93% of the data points are in zones A and B. **K-M: Ketone, lactate, and ethanol measurements for a representative subject with type 1 diabetes. K:** Ketone measurements: Beta-hydroxybutyrate measured with Yang sensor versus Menarini test strips (subject 3, same as in A-I). MAD 0.12 mM. **L:** Lactate measurements with Yang sensor versus Biosen EKF in a subject with type 1 diabetes (subject 2). MAD 0.11 mM, MARD 11.7% **M:** Ethanol measurements with Yang sensor versus Breathalyzer in a subject with type 1 diabetes (subject 4). MAD 0.018 mM.

mM showed an MAD of 0.11 mM in relation to Biosen EKF reference measurements in blood (Fig 6L). Ethanol concentrations are resolved with an accuracy of 0.02 mM MAD (Fig 6M).

## Discussion

In this early feasibility study of the YANG sensor, an investigational implantable NIR spectrometer, the sensor was found easy to implant and remove and it was well tolerated during the 28-day implantation period. Sensor implantation was done by a skilled person, in this first-in-human study a surgeon, but in the future this procedure could easily be performed by a trained endocrinologist. The implant did not give rise to any infections or wound healing issues. The current dimensions of the implantable sensor, as used in the GLOW study, as well as the implantation site in the abdominal wall, require wound closure by means of absorbable sutures (or equivalent). Future design considerations include the reduction of sensor dimensions which could allow a more practical wound closure technique.

Most study participants did not experience any inconveniences related to the implanted sensor and did not feel limited in professional or leisure activities. In this study, validated patient satisfaction questionnaires such as Diabetes Treatment Satisfaction Questionnaire (DTSQ) and Glucose Monitoring Satisfaction Survey (GMSS) were not applicable due to the use of a prototype sensor which was only active during measurement visits.

Six out of seven implanted devices transmitted good quality NIR spectra that allowed to measure glucose, ketones, ethanol, and lactate with excellent accuracy during the full implantation period. Glucose measurements by the YANG sensor did not suffer interference from paracetamol, aspartame, acetylsalicylic acid, ibuprofen, sorbitol, caffeine, fructose, and vitamin C. The confounders used in this study were selected from a list of 51 potential interferents identified through in vitro testing. These compounds show absorption in the NIR spectral range of glucose and can be expected to reach concentrations in the interstitial fluid that can interfere with the glucose prediction model unless the model is trained to take these interferents into account. Transcutaneous CGM systems with enzymatic, electrochemical based glucose measurement have been shown to be influenced by the presence of vitamin C and paracetamol (acetaminophen), as well as dopamine, maltose, xylose, mannitol [29–31]. For the Eversense implanted sensor, which measures glucose using an optical method based on the fluorescence of a glucose-indicating polymer, compounds such as tetracycline antibiotics and mannitol that absorb fluorescent light in the same wavelength range were found to act as confounders [32]. The lifetime of the implantable Eversense sensor is determined by the stability of the diboronic acid-based fluorescence-conjugated glucose indicator [12,33–35].

The innovative YANG sensor presented here is implantable, has minimal interference with daily activities and uses near-infrared spectroscopy technology, which gives the sensor a potential long lifespan and provides the capability of measuring multiple metabolites, making an evolution towards continuous multi-metabolite monitoring of diabetes possible. The lifetime of the YANG sensor is determined by the life span of its battery. The expected lifetime of the device is set to two years for regulatory purposes, but the actual battery life is potentially several years longer, as the used battery is specified to last up to 2000 charge/discharge cycles, which exceeds a two-year lifetime with daily recharge. The sensor materials in contact with the body (silicone rubber, aluminum oxide ceramic and carbon-Rich amorphous silicon) have a proven track record of stability in long-term implanted devices. Future clinical studies are planned to determine the actual safe operational lifetime of the sensor.

The limitations of the current early feasibility study are its limited size, with only seven included subjects, the relatively short implantation period of 28 days, as well as the exploratory character of the reported cross-validation for glucose, ketones, lactate and ethanol measurements. These cross-validation measurements were included as exploratory endpoints because the NIR spectra collected during this study also serve to further develop and validate the PLS models, which is why the model was trained at the start of every measurement visit.

Explorative data obtained in this study indicate that the YANG sensor can provide glucose measurements with a clinically acceptable accuracy level, as MARD values up to 10% have been accepted as the accuracy level needed for non-adjunctive use of CGM devices [36]. The overall MARD in this study was 7.1%. All MAD values in this study were also well below the clinical accuracy requirement cut-off of 15 mg/dl. In contrast with electro-chemical or biochemical measurement methods that have proportional measurement errors, the PLS regression model used by the YANG sensor expects a constant variance, resulting in a constant error band that does not increase with the measured value. This explains the low MARD values obtained with our sensor in the hyperglycemic range.

Achieving strict glycemic control without hypo- or hyperglycemic events remains challenging, as a recent international report including >520,000 children and adolescents with T1D shows that only a minority is able to achieve target HbA1c levels, with median HbA1c ranging between 7.2 to 9.4% [1].

The additional continuous measurement of ketones offers the possibility to closely monitor ketoacidosis, which is currently not part of standard diabetes monitoring, but has been strongly recommended by a 2021 consensus report [13]. This report also stresses the importance of measuring β-hydroxybutyrate in blood (or interstitial fluid) as it will provide an earlier sign of insulin deficiency than mounting glucose levels. Many situations can lead to ketosis/ketoacidosis (failure of or insufficient insulin delivery, sick days, very low carb diets, SGLT2i use, high-intensity exercise, hyperemesis gravidarum, excessive alcohol consumption), but yet many people with T1D do not even own home ketone monitoring strips, possibly due to the high cost of these strips, and in one third of cases these strips have exceeded their expiry date (16). Use of SGLT2i, especially in people with T1D, may lead to euglycemic ketoacidosis, even when administered in low doses [18,37]. In a 2019 meta-analysis, SGLT2i treatment was found to increase the risk of ketoacidosis to almost fourfold [38]. Current hybrid closed loop systems, despite being at present the best choice to achieving better glucose control, are not capable of preventing ketoacidosis in people taking SGLT2i [39,40], hence the important clinical added value of adding CKM to CGM. Albeit on rare occasions, CKM may be lifesaving.

Our sensor had a MAD of 0.12 mmol in a physiological range of 0–4 mmol hydroxybutyrate. Alva et al. tested a CKM sensor in 12 volunteers, which delivered a linear response over the 0–8 mM range with good accuracy and stability for 14 days. However, their sensor was not tested in a setting with dynamically changing blood glucose or ketone levels [41]. For another

new real-time CKM microneedle platform to monitor beta-hydroxybutyrate (BHB), only in vitro data have been reported [42].

CLM will offer the advantage of allowing persons with T1D to accurately estimate the impact of physical exertion on their glycemic control, as this depends on the relative amount of aerobic versus anaerobic exercise performed. Continuous measurement of lactate levels via an implanted YANG sensor will allow far more accurate measurement of the lactate threshold than the estimation method based on heart rate variability offered by today's high-end sports watches. In high-intensity exercise, a high lactate level itself can moreover contribute to hyperglycemia because lactate acts as an alternative substrate for glucose and provides gluconeogenic precursors for hepatic glucose production [21]. Furthermore, high levels of lactate may acutely inhibit the action of insulin on peripheral glucose uptake, an effect similar to that of the counterregulatory hormones [21,22,24]. We and others noticed that the rise in glucose is preceded by a rise in lactate [22,24].). In future automated insulin delivery systems, integration of CLM data might obviate the need of the user to announce exercise (~60 min before). Lactate monitoring could also be useful in people using metformin, as this may rarely cause lactic acidosis, mainly in conditions of acute kidney injury [26].

While less of an unmet medical need, monitoring ethanol and glucose levels concurrently will allow to provide insights to people with diabetes on the negative effect of alcohol consumption on glycemic control. Monitoring of ethanol monitoring might also potentially be useful in closed loop systems because of the potential of timely adaptation of insulin delivery by an algorithm that considers both glucose and ethanol trends.

The main impact of multi-metabolite sensing on T1D control strategies will be to reduce the burden of the disease. At present, people with T1D have to plan for and announce meals and physical activity and deal with hypoglycemic episodes. Using multi-metabolite monitoring data to adjust insulin delivery holds the promise to substantially improve the quality of life of people with diabetes. In addition to Indigo, Abbott, QuLab and the Alberta Diabetes Institute are also working on continuous multi-metabolite monitoring.

In summary, in this study, the first implantable multi-biomarker sensor, with an expected life span of at least two years, was shown to be well tolerated and produce accurate measurements of glucose, ketones, lactate, and ethanol. Next steps comprise further development of the YANG sensor from an investigational to a medical device, including an update of the sensor hardware, as well as further clinical studies to evaluate the long-term (>1 month) accuracy and performance of the sensor.

## Supporting information

**S1 Checklist. µCONSORT 2010 checklist of information to include when reporting a randomised trial\*.**
(DOCX)

**S1 File.**
(PDF)

**S2 File.**
(PDF)

**S3 File.**
(PDF)

**S4 File.**
(PDF)

## Acknowledgments

The authors acknowledge the contribution of Veerle Persy, MD, PhD (Hugin Mugin Research, Antwerp, Belgium) as an independent medical writer. Study guarantor: CDB, as principal investigator of the study.

## Author Contributions

**Conceptualization:** Gijs Klarenbeek, Danaë Delbeke, Christophe De Block.

**Data curation:** Francesca De Ridder, Rie Braspenning, Juan S. Ordonez, Patrick Lauwers, Danaë Delbeke.

**Formal analysis:** Francesca De Ridder, Juan S. Ordonez.

**Methodology:** Danaë Delbeke.

**Project administration:** Rie Braspenning.

**Supervision:** Christophe De Block.

**Validation:** Danaë Delbeke, Christophe De Block.

**Visualization:** Francesca De Ridder, Juan S. Ordonez, Gijs Klarenbeek.

**Writing – review & editing:** Francesca De Ridder, Gijs Klarenbeek, Patrick Lauwers, Kristien J. Ledeganck, Danaë Delbeke, Christophe De Block.

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
