## [Decision Letter · Decision Letter 0]

17 Aug 2023

PONE-D-23-13677Early feasibility study with an implantable near-infrared spectroscopy sensor for glucose, ketones, lactate and ethanolPLOS ONE

Dear Dr. De Block,

Thank you for submitting your manuscript to PLOS ONE. After careful consideration, we feel that it has merit but does not fully meet PLOS ONE’s publication criteria as it currently stands. Therefore, we invite you to submit a revised version of the manuscript that addresses the points raised during the review process.

We look forward to receiving your revised manuscript.

Kind regards,

Hugh Cowley

Staff Editor

PLOS ONE

Journal Requirements:

"The study was funded by Indigo Diabetes NV"

"The authors acknowledge the contribution of Veerle Persy, MD, PhD (Hugin Mugin Research, Antwerp, Belgium) as an independent medical writer. Study guarantor: CDB, as principal investigator of the study FDR received a Strategic Basic Research Grant (1S62821N) from Research Foundation – Flanders (FWO)."

"The study was funded by Indigo Diabetes NV"

"I have read the journal's policy and the authors of this manuscript have the following competing interests:

Christophe De Block: consulting fees and honoraria for speaking for Abbott, AstraZeneca, Boehringer-Ingelheim, A. Menarini Diagnostics, Eli Lilly, Medtronic, Novo Nordisk, and Roche, and research support from AstraZeneca, Boehringer-Ingelheim, Indigo Diabetes and Novo Nordisk. 

Gijs Klarenbeek (Melfin Medical Consulting) has a consultancy agreement with Indigo Diabetes.

Danaë Delbeke and Juan Ordonez are employees of Indigo Diabetes NV."

We note that one or more of the authors are employed by a commercial company: Indigo Diabetes NV.

(2) Please also provide an updated Competing Interests Statement declaring this commercial affiliation along with any other relevant declarations relating to employment, consultancy, patents, products in development, or marketed products, etc.  

Within your Competing Interests Statement, please confirm that this commercial affiliation does not alter your adherence to all PLOS ONE policies on sharing data and materials by including the following statement: "This does not alter our adherence to  PLOS ONE policies on sharing data and materials.” (as detailed online in our guide for authors http://journals.plos.org/plosone/s/competing-interests). If this adherence statement is not accurate and  there are restrictions on sharing of data and/or materials, please state these. 

Please note that we cannot proceed with consideration of your article until this information has been declared.

7. We note you have included a table to which you do not refer in the text of your manuscript. Please ensure that you refer to Table 1 in your text; if accepted, production will need this reference to link the reader to the Table.

9. We note that the original protocol file you uploaded contains a confidentiality notice indicating that the protocol may not be shared publicly or be published. Please note, however, that the PLOS Editorial Policy requires that the original protocol be published alongside your manuscript in the event of acceptance. Please note that should your paper be accepted, all content including the protocol will be published under the Creative Commons Attribution (CC BY) 4.0 license, which means that it will be freely available online, and any third party is permitted to access, download, copy, distribute, and use these materials in any way, even commercially, with proper attribution.

Therefore, we ask that you please seek permission from the study sponsor or body imposing the restriction on sharing this document to publish this protocol under CC BY 4.0 if your work is accepted. We kindly ask that you upload a formal statement signed by an institutional representative clarifying whether you will be able to comply with this policy. Additionally, please upload a clean copy of the protocol with the confidentiality notice (and any copyrighted institutional logos or signatures) removed.

10. We note that the original protocol that you have uploaded as a Supporting Information file contains an institutional logo. As this logo is likely copyrighted, we ask that you please remove it from this file and upload an updated version upon resubmission. 

**Additional Editor Comments:**

Your manuscript has been evaluated by two reviewers, and their comments are appended below and in the attached file. Please ensure you address each of the reviewers' comments when revising your manuscript.

Reviewers' comments:

Reviewer's Responses to Questions

**Comments to the Author**

1. Is the manuscript technically sound, and do the data support the conclusions?

Reviewer #1: Yes

Reviewer #2: Yes

2. Has the statistical analysis been performed appropriately and rigorously? 

Reviewer #1: Yes

Reviewer #2: Yes

3. Have the authors made all data underlying the findings in their manuscript fully available?

Reviewer #1: Yes

Reviewer #2: Yes

4. Is the manuscript presented in an intelligible fashion and written in standard English?

Reviewer #1: Yes

Reviewer #2: Yes

5. Review Comments to the Author

Reviewer #1: Authors investigated an implantable NIR sensor for glucose ketones, lactate, and alcohol. That is an important proof of concept study in a small population with T1D and healthy volunteers.

1. I would suggest adding cannabis as a reason for DKA and for ketone monitoring in introduction, that is a main issue in USA. You can find an article to cite in JAMA Internal Medicine, Diabetes Care etc.

2. You can add the high cost of blood ketone test strips compared to urine test strips in your introduction. Another legit reason and limitation.

3. line 124, 236, 299, 360, 397, 410, 417, 421, there is an error of citation, unintended words.

4. Did you use US in 4 subjects because you could not locate the sensor or just for another part of the study? Were all sensor palpable at the end of the study?

5. Is there a way to let it heal without any stiches like Eversense procedure? I understand this is a bigger sensor about 3 cm, I am asking for future practicality point.

6. How long did patients with T1D stay in hypo and hyper challenge?

7. To add interfering substance in discussion for Eversense, mannitol is also to add for interference in addition to tetracycline.

Suggestions out of the review:

1. How about using Akturk’s method (PMID: 32031415, PMID: 33543901) using NIR to find implanted sensors instead of US? You should have a plan if they get lost, especially planning a sensor for 2 years to be in.

2. What is the reason to choose the location site vs other sites like arm etc.

3. There is no consensus about what the threshold for ketone in T1D should be, different guidelines, expert opinions suggest different things just to be aware.

4. Consider high ketone levels in future studies like healthy volunteers eat strict keto and T1D using SGLT2.

Reviewer #2: Important note: This review pertains only to ‘statistical aspects’ of the study and so ‘clinical aspects’ [like medical importance, relevance of the study, ‘clinical significance and implication(s)’ of the whole study, etc.] are to be evaluated [should be assessed] separately/independently. Further please note that any ‘statistical review’ is generally done under the assumption that (such) study specific methodological [as well as execution] issues are perfectly taken care of by the investigator(s). This review is not an exception to that and so does not cover clinical aspects {however, seldom comments are made only if those issues are intimately / scientifically related & intermingle with ‘statistical aspects’ of the study}. Agreed that ‘statistical methods’ are used as just tools here, however, they are vital part of methodology [and so should be given due importance]. I look at the manuscript in/with statistical view point, other reviewer(s) look(s) at it with different angle so that in totality the review is very comprehensive. However, there should be efforts from authors side to improve (may be by taking clues from reviewer’s comments). Therefore, please do not limit the revision only (with respect) to comments made here.

COMMENTS: Although this manuscript is well drafted [and the study is excellent with respect to most of the aspects], I have few observations/concerns (different opinion) which are given below:

Agreed that this study being ‘pilot’ (feasibility study) in nature, sample size is not a big issue. However, considering the variation, sample size seems to be too small [7 participants (4 with type 1 diabetes (T1D), 3 healthy volunteers)]. I can understand that ‘Implantation’ of the sensor & six 8-hour long measurement visits are are not easy / cumbersome.

It is true [as is often quoted] that “Pilot (Proof of Concept/feasibility) studies typically involve a small number of subjects, as well as more latitude [i.e., leeway, freedom, liberty] in statistical requirements.”], in my opinion, methodological issues need to be very rigorous followed {like in case of clinical trial, CONSORT guidelines are to be strictly observed/followed}. You may definitely know that CONSORT guidelines for Pilot trial(s) is available.

Moreover, I request these learned authors to discuss the appropriateness of choice of [Healthy volunteers as] ‘control’ group and also the dis-similar handling of this group (refer to ‘Healthy volunteers’ section in lines 238 onwards) with well experienced & well qualified Biostatistician. This suggestion because I doubt about (the validity of) both these points. For ‘pilot/feasibility study’ it hardly matters / is definitely not an important issue [because even single group may suffice]. However, this knowledge will be helpful while planning a big/main trial/study.

Except these minor points, the article is acceptable. However, mind you that as pointed out in ‘important note’ above “This review pertains only to ‘statistical aspects’ of the study and so ‘clinical aspects’ should be assessed separately/independently. ‘Minor Revision’ is recommended.

6. PLOS authors have the option to publish the peer review history of their article (what does this mean?). If published, this will include your full peer review and any attached files.

Reviewer #1: No

Reviewer #2: No

---

## [Author Response · Author response to Decision Letter 0]

3 Oct 2023

Reviewer 1

1. “I would suggest adding cannabis as a reason for DKA and for ketone monitoring in introduction, that is a main issue in USA. You can find an article to cite in JAMA Internal Medicine, Diabetes Care etc”

Thank you for the suggestion, we added cannabis use to the list of risk factors for diabetic ketoacidosis, citing the findings on Kinney et al, Diabetes Care 2020 on line 78 of the revised manuscript.

2. “You can add the high cost of blood ketone test strips compared to urine test strips in your introduction. Another legit reason and limitation.”

In response to this suggestion, we updated the reasons for limited ketone testing in the discussion section of the paper (line 488 and following) as follows: “but yet many people with T1D do not even own home ketone monitoring strips, possibly due to the high cost of these strips, and in one third of cases these strips have exceeded their expiry date.”

3. “line 124, 236, 299, 360, 397, 410, 417, 421, there is an error of citation, unintended words.”

Cross-references to the tables and figures were not rendered correctly by the manuscript submission system. In the revised version, a version without field codes will be uploaded to remediate this.

4. “Did you use US in 4 subjects because you could not locate the sensor or just for another part of the study? Were all sensor palpable at the end of the study?”

The ultrasound evaluation was planned in the study design as part of the safety and biocompatibility endpoint of the study. In view of its size, there were no issues locating or retrieving the sensors at the time of explantation. A statement to this effect was added to the results section of the revised manuscript (line 335). All sensors were indeed easily palpable at the end of the implantation period.

5. “Is there a way to let it heal without any stiches like Eversense procedure? I understand this is a bigger sensor about 3 cm, I am asking for future practicality point.”

The subcutaneous layer needs to be closed with absorbable sutures. The skin wound is not a superficial small wound where wound tape, e.g. Steri-Strips (3M), could be solely used without any skin stitches. It is however a straight clean wound where wound tape could be used for wounds even over 5 cm, but they do not support lacerations well in lax skin or wounds that are under a lot of tension. Surgical glue, e.g. DERMABOND, could also be an option to close the skin. The functional tensile strength of DERMABOND is comparable to that of 5–0 sutures, however, because the first-day strength of DERMABOND is much less than that of sutures, application is limited to wounds on the face, extremities and selected areas over the torso. This modality has as disadvantage that it is brittle and subject to fracture when used on long lacerations or over skin creases. In conclusion, the wound does not have the optimal location and size to use wound tape or surgical glue, but future reduction of sensor dimensions could allow alternative wound closure techniques.

In response to this comment, the following statement was added to the discussion section of the paper (line 426 and following): “The current dimensions of the implantable sensor, as used in the GLOW study, require wound closure by means of absorbable sutures (or equivalent). Future design considerations include the reduction of sensor dimensions which could allow more practical wound closure.”

6. “How long did patients with T1D stay in hypo and hyper challenge?”

The measurement visit was planned over the full day. The patients went gradually from normoglycemia to hyperglycemia (250 or 400 mg/dl for lactate and non- lactate challenge resp), then to hypoglycemia (40 mg/dl) and back to normoglycemia state. Per protocol the max rate of change for decreasing glucose levels was 100 mg/dl/hour. 

The time spent in hypoglycaemia was documented, although it was not part of the outcome parameters of the study.

On average people spent 22 minutes in hypoglycemia (< 54 mg/dl), ranging between 9-32 minutes. When a glucose level of 40 mg/dl was reached, patients were immediately given 15 ml of a soft drink (cola).

The mean time spent in hyperglycemia (> 180 mg/dl) was 144 minutes, varying between subjects from 80 to 221 minutes. 

The mean time spent to go from the hyperglycemic maximum target (250 or 400 mg/dl for lactate and non-lactate challenge resp) to a glucose level of 40 mg/dl was 232 minutes (varying between 172-290 minutes).

7. “To add interfering substance in discussion for Eversense, mannitol is also to add for interference in addition to tetracycline.”

Mannitol was added as a potential interfering compound for the Eversense sensor on line 448 of the revised manuscript.

Suggestions

1. “How about using Akturk’s method (PMID: 32031415, PMID: 33543901) using NIR to find implanted sensors instead of US? You should have a plan if they get lost, especially planning a sensor for 2 years to be in.”

The authors thank the reviewer for this suggestion. In this study, there was no issue locating the implants. In view of the size of the implant (3 x 0.7 x 1.5 cm) and the subcutaneous implantation in the abdominal region, we do not expect any issues retrieving the sensors. It is good to know, however, that alternative localization methods exist. A statement that there were no issues locating the sensors for explantation was added to the revised manuscript (line 335).

2. “What is the reason to choose the location site vs other sites like arm etc.”

The abdominal region was chosen as the site of implantation because it provides enough space for the implantation and is easily reachable for data exchange with the current sensor and the requirements for charging the sensor and communicating with the sensor. Other implant locations can be evaluated if design aspect of the future sensor will allow. Indeed, taking into account the size of this sensor, which is a prototype, the arm would not be a suitable site.

3. “There is no consensus about what the threshold for ketone in T1D should be, different guidelines, expert opinions suggest different things just to be aware.”

Early identification of ketosis could facilitate preemptive measures to prevent impending ketoacidosis. A beta-hydroxybutyrate level < 0.6 mmol/L is generally accepted as posing no particular increased risk (cfr references listed below). For ketone levels between 0.6-1.0 mmol/L, the Australian Diabetes Educators Association (ADEA) advises that blood ketones should be checked every 2 hours, and every hour for ketone levels >1.0 mmol/L For sick day management, the American Diabetes Association (ADA) guidelines to parents of children with T1D advises to “check blood or urine ketones as often as every 4 hours,”.

• Australian Diabetes Educators Association. Clinical Guiding Principles for Sick Day Management of Adults with Type 1 and Type 2 Diabetes. Technical document. Canberra: Australian Diabetes Educators Association; 2016.

• American Diabetes Association. Checking For Ketones. Available at: http://www.diabetes.org/living-with-diabetes /treatment-and-care/blood-glucose-control/checking-for-ketones.html

• Lee MH, Paldus B, Krishnamurthy B, McAuley SA, Shah R, Jenkins AJ, et al. The Clinical Case for the Integration of a Ketone Sensor as Part of a Closed Loop Insulin Pump System. J Diabetes Sci Technol. 2019;13:967–73. https://doi.org.10.1177/1932296818822986 (cited in the manuscript as reference 14).

• Nguyen KT, Xu NY, Zhang JY, Shang T, Basu A, Bergenstal RM, et al. Continuous Ketone Monitoring Consensus Report 2021. J Diabetes Sci Technol 2021;19322968211042656. https://doi.org.10.1177/19322968211042656 (cited in the manuscript as reference 13).

4. “Consider high ketone levels in future studies like healthy volunteers eat strict keto and T1D using SGLT2.”

The increased risk of people with T1D using SGLT2i is mentioned in the discussion section of the manuscript. Since this is an early feasibility study and the sensor is currently intended for patients with diabetes, we consider measuring ketones in healthy persons following a ketogenic diet out of scope for this study. It will be considered as a potential additional application of the device in the future, however.

Reviewer 2

1. “Agreed that this study being ‘pilot’ (feasibility study) in nature, sample size is not a big issue. However, considering the variation, sample size seems to be too small [7 participants (4 with type 1 diabetes (T1D), 3 healthy volunteers)]. I can understand that ‘Implantation’ of the sensor & six 8-hour long measurement visits are not easy / cumbersome.”

Although the number of subjects is small, the number of datapoints obtained in this study is not: over the 6 measurement visits of 8 hours, the continuous measurements of the sensor were compared with values from 60-70 blood samples obtained every 5-10 minutes.

2. “It is true [as is often quoted] that “Pilot (Proof of Concept/feasibility) studies typically involve a small number of subjects, as well as more latitude [i.e., leeway, freedom, liberty] in statistical requirements.”], in my opinion, methodological issues need to be very rigorous followed {like in case of clinical trial, CONSORT guidelines are to be strictly observed/followed}. You may definitely know that CONSORT guidelines for Pilot trial(s) is available.”

The CONSORT guidelines are intended to improve the reporting of randomized trials and as such are not applicable to our study, which is not a randomized trial. The primary endpoint was to investigate the safety and biocompatibility of the sensor over a 28-day implantation period, the secondary endpoint to study the parameters of the implantation and explantation procedure, whereas the measurements of glucose, glucose, β-hydroxybutyrate, lactate and ethanol were only explorative endpoints. Larger follow-up trials taking into account statistical sample size considerations are being conducted at present.

3. “Moreover, I request these learned authors to discuss the appropriateness of choice of [Healthy volunteers as] ‘control’ group and also the dis-similar handling of this group (refer to ‘Healthy volunteers’ section in lines 238 onwards) with well experienced & well qualified Biostatistician. This suggestion because I doubt about (the validity of) both these points. For ‘pilot/feasibility study’ it hardly matters / is definitely not an important issue [because even single group may suffice]. However, this knowledge will be helpful while planning a big/main trial/study.”

It was not the objective of this study to “compare” persons with T1D and healthy volunteers. The reason not to perform this early feasibility study only in healthy volunteers was to already obtain safety data in persons with T1D, and to allow for measurements of glucose across the whole relevant clinical range, which would not be possible in healthy volunteers.

---

## [Decision Letter · Decision Letter 1]

17 Oct 2023

PONE-D-23-13677R1Early feasibility study with an implantable near-infrared spectroscopy sensor for glucose, ketones, lactate and ethanolPLOS ONE

Dear Dr. De Block,

Thank you for submitting your manuscript to PLOS ONE. After careful consideration, we feel that it has merit but does not fully meet PLOS ONE’s publication criteria as it currently stands. Therefore, we invite you to submit a revised version of the manuscript that addresses the points raised during the review process.

Please submit your revised manuscript by Dec 01 2023 11:59PM. If you will need more time than this to complete your revisions, please reply to this message or contact the journal office at plosone@plos.org. Please include the following items when submitting your revised manuscript:A rebuttal letter that responds to each point raised by the academic editor and reviewer(s). You should upload this letter as a separate file labeled 'Response to Reviewers'.A marked-up copy of your manuscript that highlights changes made to the original version. You should upload this as a separate file labeled 'Revised Manuscript with Track Changes'.An unmarked version of your revised paper without tracked changes. You should upload this as a separate file labeled 'Manuscript'.

We look forward to receiving your revised manuscript.

Kind regards,

Dured Dardari, Ph.D

Academic Editor

PLOS ONE

Reviewers' comments:

Reviewer's Responses to Questions

**Comments to the Author**

1. If the authors have adequately addressed your comments raised in a previous round of review and you feel that this manuscript is now acceptable for publication, you may indicate that here to bypass the “Comments to the Author” section, enter your conflict of interest statement in the “Confidential to Editor” section, and submit your "Accept" recommendation.

Reviewer #1: All comments have been addressed

Reviewer #2: (No Response)

2. Is the manuscript technically sound, and do the data support the conclusions?

Reviewer #1: Yes

Reviewer #2: (No Response)

3. Has the statistical analysis been performed appropriately and rigorously? 

Reviewer #1: Yes

Reviewer #2: (No Response)

4. Have the authors made all data underlying the findings in their manuscript fully available?

Reviewer #1: Yes

Reviewer #2: (No Response)

5. Is the manuscript presented in an intelligible fashion and written in standard English?

Reviewer #1: Yes

Reviewer #2: (No Response)

6. Review Comments to the Author

Reviewer #1: (No Response)

Reviewer #2: COMMENTS: Few of the comments made on first draft were addressed, however, frankly speaking, I am/was not very much convinced (or happy) for reasons given or arguments made.

While answering my second comment you said “The CONSORT guidelines are intended to improve the reporting of randomized trials and as such are not applicable to our study, which is not a randomized trial” which is surprising. As you know, all ‘Clinical Trials’ [random or non-random] must follow Consolidated Standards of Reporting Trials (CONSORT) guidelines [letter ‘R’ in CONSORT stands for ‘Reporting’ and not for ‘Random’]. Since your article type is ‘Clinical Trial’, you are supposed to cover at least important items in CONSORT in the report (which is not seen). You may be aware of one article on guidelines for reporting non-randomised studies {Reeves BC, Gaus W. ‘Guidelines for reporting non-randomised studies’, Forsch Komplementarmed Klass Naturheilkd 2004 Aug;11 Suppl 1:46-52. doi: 10.1159/000080576}. Non-randomised studies (NRSs) are more susceptible to bias. CONSORT has improved the reporting of key information, highlighting missing key information for users. Researchers have a responsibility to report essential information that allows users to assess the susceptibility of NRS to selection, performance, detection and attrition bias.

I feel, ‘let the respected editor decide the future course’ and I do not have any specific recommendation as such [though only as system requirement, I choose reject].

7. PLOS authors have the option to publish the peer review history of their article (what does this mean?). If published, this will include your full peer review and any attached files.

Reviewer #1: No

Reviewer #2: No

---

## [Author Response · Author response to Decision Letter 1]

10 Nov 2023

Reviewer 1: no comments

Anwer to the comments of Reviewer 2:

We agree with the reviewer that rigorous reporting methodology is of key importance. Please find the filled out CONSORT checklist below (This information is also uploaded as a supplemental information file in the editorial manager application). 

We wish to additionally point out that the PlosOne Editorial Office requested us to provide a filled out CONSORT or TREND reporting checklist. The latter we already added to our submission in May 2023. The TREND checklist has the same purpose and contains more or less the same information as the CONSORT checklist but is intended for non-randomized studies.

Please also take note of the fact that the Protocol and funding are not included in the manuscript as per the Journal’s instructions to authors. These elements are required for submission, however, and have been provided by us in the journal’s submission system. Both the funding statement and the study protocol will be published by PlosOne alongside the manuscript.

---

## [Decision Letter · Decision Letter 2]

4 Mar 2024

Early feasibility study with an implantable near-infrared spectroscopy sensor for glucose, ketones, lactate and ethanol

PONE-D-23-13677R2

Dear Dr. Christophe De Block

We’re pleased to inform you that your manuscript has been judged scientifically suitable for publication and will be formally accepted for publication once it meets all outstanding technical requirements. 

I would like to apologise for the delay in this decision but I was compelled to do 3 rounds of review and I have decided that no more  revisions are required before publication

Kind regards,

Dured Dardari, Ph.D

Academic Editor

PLOS ONE

Additional Editor Comments (optional):

Reviewers' comments:

Reviewer's Responses to Questions

**Comments to the Author**

1. If the authors have adequately addressed your comments raised in a previous round of review and you feel that this manuscript is now acceptable for publication, you may indicate that here to bypass the “Comments to the Author” section, enter your conflict of interest statement in the “Confidential to Editor” section, and submit your "Accept" recommendation.

Reviewer #3: All comments have been addressed

2. Is the manuscript technically sound, and do the data support the conclusions?

Reviewer #3: Yes

3. Has the statistical analysis been performed appropriately and rigorously? 

Reviewer #3: Yes

4. Have the authors made all data underlying the findings in their manuscript fully available?

Reviewer #3: Yes

5. Is the manuscript presented in an intelligible fashion and written in standard English?

Reviewer #3: No

6. Review Comments to the Author

Reviewer #3: Dear Authors,

This manuscript addresses a crucial subject, considering the importance of monitoring to reduce diabetes complications and the lack of implanted sensors to measure glucose, ketones, lactate and alcohol. The authors could address the comments of the previous reviewer, but a few corrections to the writing in the whole manuscript need to be made. The abstract says the lactate was measured with GlucoMen LX Plus, and the item Blood Sample Analyses says it was measured using EKF Diagnostic-Biosen C-Line Glucose and Lactate Analyser. It is essential to evaluate whether it confirms the accuracy and safety with a larger population and, for a more extended period, to evaluate if the encapsulation may interfere with the results.

Yours Sincerely

7. PLOS authors have the option to publish the peer review history of their article (what does this mean?). If published, this will include your full peer review and any attached files.

Reviewer #3: No
